# A Review of the Significance in Measuring Preoperative and Postoperative Carcinoembryonic Antigen (CEA) Values in Patients with Medullary Thyroid Carcinoma (MTC)

**DOI:** 10.3390/medicina57060609

**Published:** 2021-06-11

**Authors:** Ioannis Passos, Elisavet Stefanidou, Soultana Meditskou-Eythymiadou, Maria Mironidou-Tzouveleki, Vasiliki Manaki, Vasiliki Magra, Styliani Laskou, Stylianos Mantalovas, Stelian Pantea, Isaak Kesisoglou, Konstantinos Sapalidis

**Affiliations:** 11st Department of Surgery, “Papageorgiou” General Hospital of Thessaloniki, Aristotle University of Thessaloniki, 54621 Thessaloniki, Greece; ioannispassos@gmail.com; 2Department of Neurosurgery, 424 General Military Hospital of Thessaloniki, Aristotle University of Thessaloniki, 54621 Thessaloniki, Greece; lisa.stefanidou@gmail.com; 3Department of Histology and Embryology, Faculty of Medicine, School of Health Sciences, Aristotle University of Thessaloniki, 54621 Thessaloniki, Greece; meditskou@gmail.com; 4Department of Pharmacology, Faculty of Medicine, School of Health Sciences, Aristotle University of Thessaloniki, 54621 Thessaloniki, Greece; mmyronidauth@gmail.com; 53rd University Surgical Department, “AHEPA” University Hospital of Thessaloniki, Aristotle University of Thessaloniki, 54621 Thessaloniki, Greece; vassiamanaki@gmail.com (V.M.); valia.magra@gmail.com (V.M.); stelaskou@gmail.com (S.L.); steliosmantalobas@yahoo.gr (S.M.); ikesis@auth.gr (I.K.); 6UMF “Victor Babes”, Surgery Clinic II, 300041 Timisoara, Romania

**Keywords:** medullary thyroid carcinoma (MTC), carcinoembryonic antigen (CEA), preoperative, embryologic origin, monoclonal antibodies

## Abstract

*Background and Objectives*: Medullary thyroid carcinoma (MTC) accounts for 1–2% of all thyroid malignancies, and it originates from parafollicular “C” cells. Carcinoembryonic antigen (CEA) is a tumor marker, mainly for gastrointestinal malignancies. There are references in literature where elevated CEA levels may be the first finding in MTC. The aim of this study is to determine the importance of measuring preoperative and postoperative CEA values in patients with MTC and to define the clinical significance of the correlation between CEA and the origin of C cells. *Materials and Methods*: The existing and relevant literature was reviewed by searching for articles and specific keywords in the scientific databases of PubMedCentraland Google Scholar (till December 2020). *Results*: CEA has found its place, especially at the preoperative level, in the diagnostic approach of MTC. Preoperative CEA values >30 ng/mL indicate extra-thyroid disease, while CEA values >100 ng/mL are associated with lymph node involvement and distant metastases. The increase in CEA values preoperatively is associated with larger size of primary tumor, presence of lymph nodes, distant metastases and a poorer prognosis. The clinical significance of CEA values for the surgeon is the optimal planning of surgical treatment. In the recent literature, C cells seem to originate from the endoderm of the primitive anterior gut at the ultimobranchial bodies’ level. *Conclusions*: Although CEA is not a specific biomarker of the disease in MTC, itsmeasurement is useful in assessing the progression of the disease. The embryonic origin of C cells could explain the increased CEA values in MTC.

## 1. Introduction

Medullary thyroid carcinoma (MTC) accounts for 1–2% of all thyroid carcinomas. This malignancy originates from parafollicular cells, or “C” cells, which produce calcitonin [1]. MTC is more common in its sporadic form (80%); however, it is less common as an autosomal dominant inherited disorder, such as “Multiple Endocrine Endoplasia” MEN2A, MEN2B syndromes, and familial medullary carcinoma of the thyroid (FMTC). FMTC is a MEN2A typethat incorporates MTC, but not the other MEN2A characteristics. The MTC of MEN2A usually has a better prognosis than the MTC of MEN2B or the sporadic MTC.A patient with sporadic MTC typically presents a painless palpable thyroid mass, which isusually diagnosed by FNA and immunohistochemistry, or elevated calcitonin levels. Carcinoembryonic antigen (CEA) may also be found elevated in MTC [2].

Nodal metastasis affects 75% of patients, with the majority of cases affecting the central compartment, ipsilateral and contralateral jugulocarotid chains. Treatment of MTC is usually surgical, as C cells do not respond to iodine therapy, and includes total thyroidectomy with central cervical lymph node dissection and, if indicated, extension of the lymph node dissection to the lateral cervical compartments [3]. Most patients with MTC or a “genetic” predisposition to develop MTC should undergo at least total thyroidectomy. Patients with MEN2B syndrome and mutation in the “rearranged during transfection” RET gene should undergo prophylactic total thyroidectomy within the first year of life or at the time of diagnosis [4,5]. Lymph node clearance at level VI may be omitted in patients younger than 1 year old with MEN2B syndrome and in patients younger than 5 years old with MEN2A and FMTC undergoing total prophylactic thyroidectomy, unless there are nodules larger than 5 mm, increased calcitonin levels, or signs of lymph node metastasis [6,7].

Carcinoembryonic antigen (CEA) is perhaps the most widely studied tumor marker, originally detected by Gold et al. in 1965. It is an onco-fetal protein that is normally detected in fetal life, but can also be found in low concentrations in healthy adults. Structurally, it is a glycoprotein with a molecular weight of 200 kDa and is a component of the glycocalyx, on the endo-luminal side of the cell membrane of normal epithelial intersitial cells. CEA belongs to the super-family of immunoglobulins called the family of CEA-related cell adhesion molecules (CEACAMs). These include CEACAM1, CEACAM3, CEACAM4, CEACAM5 (CEA), CEACAM6, CEACAM7 and CEACAM8. This molecule is secreted into the circulation and can be found in the mucous secretions of the stomach, small intestine and bile duct. It is also found in the serum of patients with colorectal cancer, as well as in cancer of the stomach, pancreas, lungs and breast. Although its exact function is unknown, CEA has been shown to be involved in cell adhesion and so it may inhibit apoptosis [8].

Serum CEA levels below 2.5 ng/mL are normal, 2.5–5.0 mg/mL are considered marginal, while values above 5 ng/mL are considered elevated. Values around the upper limit may appear in benign diseases such as inflammatory bowel disease, pancreatitis, liver cirrhosis, endometriosis, autoimmune diseases and chronic obstructive pulmonary disease. Smoking can also increase CEA, so the upper normal limit for smokers is considered to be 5 ng/mL. However, CEA is not useful as a screening test because of its low sensitivity at the early stage of the disease. Elevated CEA levels occur in a 5–40% of patients with localized disease [8,9].

However, there are reports in the literature where elevated CEA levels without other clinical findings may be the first and unique finding in MTC [10].

The correlation between high CEA values and MTC, as well as the usefulness of CEA in diagnosing and monitoring patients with MTC, is not new. In a 1976 study, Ishikawa and Hamada observed high CEA values in patients diagnosed postoperatively with MTC [11]. In recent years, there are more studies trying to integrate CEA into both preoperative diagnosis and postoperative follow-up of patients with medullary thyroid carcinoma, particularly in combination with the measurement of calcitonin, to define its prognosis.

The purpose of this study is to determine the clinical importance of measuring preoperative and postoperative CEA values in patients with medullary thyroid carcinoma by studying the current literature, and to define the significance of the correlation between CEA and the origin of C cells, as well as to find possible new therapeutic interventions and targets using as a “tool” the detection of CEA in the body of patients with MTC.

## 2. Materials and Methods

In order to write this manuscript, the existing and relevant literature was reviewed by searching for articles in the scientific databases of PubMed, Central, and Google Scholar. Only English articles were included. The keywords used were: medullary thyroid cancer (MTC), carcinoembryonic antigen (CEA), parafollicular or C cells, embryological origin, anti-CEA mAbsand guidelines. Then, the articles that clearly described the correlation (positive or negative) of preoperative or postoperative CEA values with the stage and prognosis of MTC were selected, as well as those articles (all types of articles) that commented about targeted therapies based on the above correlation. All types of articles were selected, including case reports due to the lack of reviews. In addition, articles that correlated the expression of CEA with the new data on embryonic C cell maturation and MTC development were also selected. Their initial classification was based on the date of the article publication and then on the degree of relevance to the logical continuity of the text (Table A1).

## 3. Discussion

The correlation between high CEA values and MTC, as well as the usefulness of CEA in diagnosing and monitoring patients with MTC, is not new. In a 1976 study, Ishikawa and Hamada observed high CEA values in patients diagnosed postoperatively with MTC, and they even made an initial correlation of the slow decline in CEA in patients with advanced disease [11]. However, the most impressive fact is that they tried to give an explanation about the increase in CEA values in patients with MTC, unlike other thyroid tumors. Similar observations concerning the increase in CEA values in MTC were made by Wells and Mendelssohn in 1978 and 1984, respectively [12,13].

In order to understand the association between CEA and MTC, the new data on the embryological origin of parafollicular C cells and how they are related to the development of MTC should be used as established knowledge [14,15].

The thyroid gland contains a specific population of hormone-producing cells, which are called, as already mentioned, parafollicular cells or “C” cells. These cells produce calcitonin, ahypocalcemic hormone that acts as a natural antagonist of parathormone. In addition, the thyroid gland includes a rich network of capillaries that surround each follicle and provide a systematic distribution of hormones that are released. The extra-mesenchymal fibroblasts originating from the neural crest form the stromal compartment, which encases and separates the follicular thyroid tissue [16]. The thyroid gland also includes other intermediate cells such as macrophages and mast cells, which have attracted scientific interest due to their functions in thyroid carcinoma [17,18].

By definition, and with the enigma of the origin of C cells remaining, neuroendocrine cells are those that receive nerve impulses and secrete signals in response (originally classified as APUD cells), and are characterized by the uptake of precursor forms of amines and decarboxylation, a feature they share with neurons. Consequently, it was not difficult for the scientific community to adopt the idea in the early 1970s that the neural crest, from the neural tube, is the potential source of all neuroendocrine cells. However, these theories proved to be based on misconceptions, and the hypothesis of neural crest was eventually abandoned, as the absorptive intestinal epithelium and intestinal neuroendocrine cells appeared (by genetic tracing techniques) to be differentiated from the same precursors, originating from the endoderm. However, thyroid C cells are an exception to the rule. Linear genetic tracing in mice has now shown that neuroendocrine cells in the thyroid gland are also of endodermal origin, suggesting that ultimobranchial bodies not only act as precursors of “C” cells, but they are also their true embryonic origin. This also suggests that calcitonin-producing cells, which are found in the lower vertebrates, may also originate from the anterior endoderm. A unified origin of follicular thyroid cells and C cells, although from different areas of the endoderm, can answer questions about the histogenesis of mixed thyroid cancers that were difficult to explain. The discovery that thyroid C cells originate from the endoderm opens new perspectives in the search for potential therapeutic targets in the treatment of C cell-derived neoplasms, which are characterized by high infiltration [17,18]. This discovery also leads to the need to reclassify MTC in the family of endodermal neuroendocrine tumors. Compatible with key roles in the development of neuroendocrine tumors, MTC C cells differentially express the transcription factors Foxa1 and Foxa2, which are associated with tumor growth. This partly summarizes the morphogenetic pattern of Foxa1 and Foxa2 observed during embryonic development of C cell precursors [19].

Back to the original hypothesis of the correlation between CEA and MTC, studies at cellular level, as well as at the level of mRNA expression, have correlated MTC with increased CEA expression. In an experimental study on various types of cancercell lines, Wakabayashi et al. in 2014 investigated the mRNA expression profile of members of the CEACAM family in the cancer cell lines, which included thyroid cancer. It has been observed that CEACAMs, with the exception of CEACAM8, are expressed in TT cell lines, which are MTC cells, whereas CEACAM4 is specifically expressed in this cell line alone. Specific expression of CEACAM4 in MTC cell lines could therefore potentially differentiate MTC from other CEA-secreting tumors [20].

CEA levels of approximately 2–4 ng/mL are found in healthy individuals. Levels more than 10 ng/mL are usually associated with malignant diseases, such as colorectalcarcinoma, ovarian cancer, breast cancer, thyroid carcinoma, non-small cell lung cancer, and appendix mucinous cystadenoma. Non-neoplastic conditions may also occur. These include heavy smoking, gastric ulcers, cholecystitis, liver cirrhosis, pancreatitis, inflammatory bowel disease, and orlistat intake.

According to the Revised Guidelines of the American Thyroid Association for the Management of Medullary Thyroid Carcinoma, which were published in 2015 in Thyroid, CEA is not a specific biomarker for MTC, so false-positive CEA levels should be highlysuspected. Differential diagnosis is required in such cases. However, determination of serum CEA levels is useful in assessing disease progression in patients with clinically evident MTC, but also in monitoring patients after thyroidectomy. In patients with MTC, the concomitant increase in CEA and calcitonin levels indicates disease progression. Basic CEA and calcitonin levels should be calculated simultaneously. In patients with advanced MTC, a notable increase in CEA disproportionate to low serum calcitonin levels, but also the finding of normal or low CEA and calcitonin levels indicate poorly differentiated MTC [21].

Other scientificsocieties have included the measurement of CEA as an adjunct to the diagnosis and monitoring of patients with MTC. Chen et al. published in 2010 the guidelines of the North American Society of Neuroendocrine Tumors for the diagnosis and treatment of MTC. According to them, CEA can be used as a biomarker of the specific disease and which selectively tends to be selectively expressed in less differentiated tumors. In particular, preoperative CEA values above 30 ng/mL indicate extra-thyroid expansion of the disease, while values above 100 ng/mL are associated with extensive lymph node involvement and distant metastases [22].

More recently, the measurement of CEA has been included in the UK national guidelines for the diagnostic and therapeutic approach to MTC. These guidelines, published in 2016, include the measurement of CEA preoperatively in patients with suspected or established MTC, as well as postoperatively, as part of routine monitoring of the disease [23].

The possibility of using CEA as a cancer indicator to predict the severity of the disease has been studied in a retrospective study published in 2018, where preoperative and postoperative calcitonin and CEA values were measured in 33 patients who underwent total thyroidectomy with central and/or lateral lymph node dissection for MTC during the period 2003–2016. Preoperative CEA values have been found to correlate with disease extent and range of surgical resection, but CEA is a better indicator of tumor aggression and advanced disease. Additionally, CEA values can relatively reliably predict tumor size, the presence of lymph node metastases in the central cervical region, and MTC mortality. More specifically, CEA values >271 ng/mL were significant for larger primary tumor size, CEA values >377 ng/mL were positively correlated with more advanced stages of the disease, with lymph node metastases in the lateral cervical compartments, and with reduced chances of biochemical cure (values >405 mg/mL). Turkdogan et al. suggest that CEA levels >500 ng/mL are a good marker for advanced disease and MTC mortality of up to 67%, while being more cost-effective than calcitonin and probably other markers [24].

Carcinoembryonic antigen seems to have found its place, especially at the preoperative level, in the diagnostic approach of MTC [25,26,27]. Nien et al. suggest that, in the case of elevated CEA values in “routine” biochemical testing, further investigation should focus not only on upper and lower gastrointestinal endoscopy to rule out evidence of gastrointestinal malignancy, but also on neck ultrasound and measuring serum calcitoninto find a possible MTC. In addition, at the postoperative level, CEA values fall more slowly than calcitonin, making the latter a more reliable biochemical indicator in predicting biochemical cure [28].

In a retrospective study published in 2007, Machens and Dralleused multivariate analysis in 150 patients preoperatively and studied specific biomarkers including CEA in the treatment of MTC, concluding that elevated CEA levels indicate advanced disease, larger primary tumor size, possible extra-thyroid expansion and distant metastases. More specifically, CEA values >30 ng/mL indicate the presence of lymph node metastases in the central and ipsilateral cervical compartment in approximately 70%. CEA values >100 ng/mL indicate unilateral cervical lymph node metastases and distant disease in percentages reaching 90% and 75%, respectively. The authors conclude that preoperative CEA values can be used reliably in the design of surgery and concomitant lymphadenectomy at MTC [29,30].

CEA values can therefore be taken into account as risk factors for lymph node metastases in the central and lateral cervical compartment and thus direct the extent of the surgical procedure. In a clinical study published in 2018, Fan et al. studied the risk factors for cervical lymph node metastases in 65 patients with histologically documented MTC. These factors included tumor size (>1 cm), multifocal disease, and tumor infiltration of the thyroid capsule. In this study, preoperative CEA values ≥30 ng/mL were statistically significant for the presence of lymph node metastases in the central cervical compartment relative to CEA values <30 ng/mL. They conclude that prophylactic lateral cervical lymph node dissection should be performed in patients with MTC and capsule infiltration, as well as at high preoperative CEA values [31].

Van Veelen and De Groot in 2009 confirm the above findings, as CEA, although a non-specific biological indicator in the diagnosis of MTC, can be associated preoperatively with tumor size, MTC recurrence, disease prognosis and presence of lymph node metastases. According to them, CEA is a more reliable indicator of the extent of the disease preoperatively than in the postoperative follow-up of patients, in which CEA values may be within normal levels despite the presence of residual disease. In rare cases of MTC that do not secrete calcitonin, CEA becomes more important. They conclude that plasma CEA has a lower sensitivity and specificity than plasma calcitonin in the diagnostic approach of MTC, but may be a reliable preoperative prognostic indicator for the disease [32].

A review by Thomas and Goldstein published in 2019 is in the same direction. According to them, CEA, although a non-specific diagnostic indicator, can be relatively reliably correlated with the prognosis of the disease, especially at the preoperative level. The authors also report that calcitonin and CEA secretion are proportional to total C cell mass [33].

In a retrospective study published in 2020, Zheng-Pywell et al. examined 88 patients with MTC diagnosis in the period 2008–2014. The study was initially based on data that found that elevated CEA values were indicative of disease aggressiveness, associated with larger primary tumors, positive lymph nodes, distant metastases, decreased biochemical cure rates and increased mortality. The authors conclude that CEA should be evaluated in combination with calcitonin both preoperatively and postoperatively in order to decide about further imaging investigation of possible metastatic disease. The lack of harmonization of CEA and calcitonin values, especially postoperatively, is an indication of metastatic disease, given the prognostic value of the above biomarkers. However, they point out that CEA cut-off values cannot be accurately determined, which could differentiate an aggressive locally advanced from a metastatic disease [34].

The doubling times of both CEA and calcitonin are very important indicators of prognosis and disease progressionin MTC. In a study published in 2008 inEuropean Journal of Endocrinology, Giraudet et al. studied the doubling times of CEA and calcitonin in patients with diagnosed and advanced MTC. They found thatdoubling time values could be used to assess the disease progression and overall prognosis [35]. The above findings are consistent with a previous retrospective study of 65 patients who underwent total thyroidectomy and bilateral cervical lymph node dissection for MTC and who underwent postoperative follow-up that included consecutive measurements of CEA and calcitonin values from 6 months to 30 years postoperatively. In this cohort of Barbet et al., it was shown that the doubling time of calcitonin and CEA values could be used to estimate the prognosis of the disease [36]. Even in the case of a cytoreductive surgical operation, an increase of the doubling times of the above biomarkers as well as a slower progression of the disease have been shown [37].

Persistent rise in CEA values after total thyroidectomy and bilateral central cervical lymph node dissection for MTC is an indication of metastatic disease, even in the absence of increased calcitonin values. In a recent case report published recently, an increase in CEA values postoperatively with a parallel normalization of postoperative calcitonin values revealed the presence of bone metastasis in the left scapula [38].

In contrast to the above studies, Yip et al. published in 2011 the results of a retrospective study, which was conducted for the time period 1980–2009 in 104 patients treated for MTC. Their findings showed that preoperative measurement of calcitonin, not CEA, reflects the extent of the disease, a finding that supports the use of preoperative calcitonin values as an indicator of tumor size and presence of lymph node metastases. They also reported that postoperative measurement of calcitonin and CEA cannot be correlated with the extent and radicalness of thyroidectomy performed [39].

Rarely, patients are diagnosed with MTC without an increase in calcitonin and CEA values, even in the case of advanced disease [40,41,42]. In 2019, Gambardella et al. published the results of a systematic review involving patients with non-secretory MTC. They included patients with concomitant lymphadenopathy or metastatic disease. They concluded that absolute calcitonin and CEA values decrease the possibility of differentiation between loco-regional and metastatic disease [43,44].

Therapiesbased on monoclonal antibody (mAb-based therapies) have recently played an important role in cancer treatment. When labeled with radionuclides, monoclonal antibodies (mAbs) represent promising tools for diagnostic or even therapeutic purposes. Nowadays, the most accurate imaging is achieved with the use of positron emission tomography (PET). Immunotherapy using PET (immuno-PET), which combines the high sensitivity and analysis of a PET camera with the specificity of a monoclonal antibody, is the most representative approach of these new imaging and treatment methods. Like radio-immunotherapy (RIT), PET immunotherapy can be performed using direct or pre-targeted radiolabeled monoclonal antibodies to enhance contrast and imaging accuracy. Pre-targeted immuno-PET has been developed against various antigens with promising results reported in CEA-expressing tumors (CEA or CEACAM5) using a specific mAb and a radiolabeled peptide [45]. Radiolabeled anti-CEA mAbs are excellent agents for the imaging of recurrent, residual or metastatic MTC, especially in terms of staging and monitoring the disease progression [46].

MTC, which is characterized by strong CEA expression, and represents a related cancer model for pre-targeted immuno-PET. A recent and innovative clinical study, published in 2019 by Bodet-Milin et al., reported high tumor uptake and increased contrast imaging using pre-targeted anti-CEA immuno-PET in patients with metastatic MTC [47,48].

Pretargeted radioimmunotherapy (PRAIT) consists of the administration of a bispecific monoclonal antibody and then, after a few days, of a radiolabeled divalent hapten. The radioactive hapten binds to the monoclonal antibody that is already bound to targeted CEA-producing cancer cells. This method was studied in 18 patients with stage IV metastatic MTC, and the results showed control of disease progression in 76% of patients, with a mean PFS at 13.6 months and OS at 43.9 months. The doubling time values of biomarkers (calcitonin and CEA) was increased (extended) by 100%, which was associated with a clinical response and better OS. The above researchers concluded that measurement of biomarkers can reliably predict the clinical response of advanced MTC to systemic therapy [49].

The importance of measuring CEA in MTC is also shown in several case reports. In 2017, Chen et al. described the case of persistently high CEA values postoperatively in a patient treated for colorectal cancer and in whom 18F-FDG-PET/CT (fluorodeoxyglucose (FDG)-positron emission tomography) was clear with the exception of increased thyroid uptake. MTC was diagnosed and treated surgically [50]. Others recommend the investigation of tumors of neuroendocrine origin and especially MTC, in case of elevated CEA values [51,52].

## 4. Conclusions

Regarding the measurement of CEA values in patients with MTC, it could be concluded that (Table A2):The correlation between the increase in CEA values and MTC is not a new observation.New data concerning the embryonic origin of C cells (from which MTC is derived) from the endoderm of the primary anterior intestine and the ultimobranchial bodies, make MTC a neuroendocrine tumor.CEA is not a specific biomarker of the disease in MTC, but its measurement is useful in assessing the progression of the disease, before and after thyroidectomy.The increase in postoperative CEA values, which is not accompanied by a corresponding change in calcitonin values, is an indication of poorly differentiated MTC, disease progression and poor prognosis.Preoperative CEA values >30 ng/mL indicate extra-thyroid disease, while CEA values >100 ng/mL are associated with lymph node involvement and distant metastases.Postoperative measurement of CEA in the follow-up of patients with MTC has become routine.The increase in CEA values preoperatively has a positive correlation with a larger size of primary tumor, with the presence of lymph nodes and distant metastases, but also with a poorer prognosis.The clinical significance of CEA values for the surgeon is the optimal planning of surgical treatment and the extent of resection (total thyroidectomy + central cervical lymph node dissection + unilateral lateral cervical lymph node dissection when CEA >30 mg).Simultaneous measurement of calcitonin values is important, especially in cases of MTCs that do not secrete CEA, while their doubling time values may be used as an indication of disease progression.Treatment with radiolabeled monoclonal antibodies against CEA (anti-CEA mAbs, pre-targeted radio-immunotherapy), with the help of positron emission tomography (PET) imaging, is the promising future in the diagnosis and treatment of metastatic MTC.Persistent elevated CEA values after colon cancer surgery for and negative diagnostic workup for metastatic disease should raise a high clinical suspicion of developing MTC.

## Data Availability

Not applicable.

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
