# Peer review of "A Review of the Significance in Measuring Preoperative and Postoperative Carcinoembryonic Antigen (CEA) Values in Patients with Medullary Thyroid Carcinoma (MTC)"

_medicina, 2021, doi:10.3390/medicina57060609_

Round 1

Reviewer 1 Report

The authors present a rather descriptive review of the literature regarding CEA as a diagnostic/prognostic marker for MTC - with no attempt to meta-analyse the data presented in recent publications.  Whilst  this leaves the conclusion somewhat subjective it is a useful review of current opinions in this field.

I have a few minor comments

174 - not sure 'companies' is the correct term here

200 need to expand on ref 24,  whilst CEA is cheaper and possibly a good marker for advance disease – I don’t think the conclusion of ref 24 is that CEA necessarily has a better cost-effectiveness

353 - don't understand term ‘loco-regional’

320 - rephrase 'could be said' which is ambiguous

the document has many missing spaces between words

Author Response

Thank you very much for your interesting and useful comments regarding our manuscript. 

174- companies --> societes 

200- sentence changed

353- locally advanced

320- sentence changed

missing spaces between words corrected

Reviewer 2 Report

Interesting and comprehensive review of the role of CEA levels in the management of Medullary Thyroid Carcinomas in addition to other more "traditional" parameters. Easy-to-read work with an essential story on biomarkers' findings on various forms of MTC. Abundant and updated references. The most interesting part are undoubtedly the schematic conclusions that offer a precious mnemonic framework for anyone who needs to face the diagnosis and follow up of these endocrine neoplasms. There are some grammatical inaccuracies and small text errors (spaces, returns, commas). Ie lines 26, 29, 32, 34, 49, 53 ....

Author Response

Thank you very much for your useful comments. We made all the suggested alterations.

Reviewer 3 Report

This manuscript is more of a book report than a scientific presentation. While many relevant citations have been pulled from the literature there is no specific data cited, analyzed or scientifically evaluated. Since the authors report no personal, new or novel data the forward looking conclusions are challenged at best.

Author Response

Response to the reviewer:

    Thank you for your comments. We took into consideration your opinion concerning our manuscript. We have still revised the manuscript for the other two reviewers and there was not a comment concerning the conclusions. As you could imagine we cannot change the structure from the beginning. It would be very useful for us if you could provide us with more specific suggestions.

    Thank you in advance.

Round 2

Reviewer 3 Report

No significant changes in revision.

Author Response

Thank you very much for your reply and your useful help. 
